# Mitochondrial Dynamics in Pulmonary Hypertension

**DOI:** 10.3390/biomedicines12010053

**Published:** 2023-12-25

**Authors:** Ed Wilson Santos, Subika Khatoon, Annarita Di Mise, Yun-Min Zheng, Yong-Xiao Wang

**Affiliations:** 1Department of Molecular and Cellular Physiology, Albany Medical College, Albany, NY 12208, USA; cavalce@amc.edu (E.W.S.); khatoos@amc.edu (S.K.); annarita.dimise@uniba.it (A.D.M.); 2Department of Biosciences, Biotechnologies and Biopharmaceutics, University of Bari, Via Orabona, 4, 70125 Bari, Italy

**Keywords:** mitochondria, circadian molecules, nicotine, hypoxia, fusion, fission, pulmonary vascular dysfunction

## Abstract

Mitochondria are essential organelles for energy production, calcium homeostasis, redox signaling, and other cellular responses involved in pulmonary vascular biology and disease processes. Mitochondrial homeostasis depends on a balance in mitochondrial fusion and fission (dynamics). Mitochondrial dynamics are regulated by a viable circadian clock. Hypoxia and nicotine exposure can cause dysfunctions in mitochondrial dynamics, increases in mitochondrial reactive oxygen species generation and calcium concentration, and decreases in ATP production. These mitochondrial changes contribute significantly to pulmonary vascular oxidative stress, inflammatory responses, contractile dysfunction, pathologic remodeling, and eventually pulmonary hypertension. In this review article, therefore, we primarily summarize recent advances in basic, translational, and clinical studies of circadian roles in mitochondrial metabolism in the pulmonary vasculature. This knowledge may not only be crucial to fully understanding the development of pulmonary hypertension, but also greatly help to create new therapeutic strategies for treating this devastating disease and other related pulmonary disorders.

## 1. Introduction

Mitochondria have for a long time been identified as being the “powerhouse” of the cell, where the production of energy provides the entire functioning of cellular metabolism. However, mitochondria are much more than that, they can act in several other biological processes. In the last few decades, research advancements have revolutionized our comprehension of multifaceted and intricate mitochondrial functions. These complex organelles have emerged as central players in diverse biological processes, including redox signaling, calcium homeostasis, apoptosis, autophagy, and other cellular responses. Moreover, cutting-edge imaging techniques have facilitated a thorough investigation of mitochondrial dynamics, which control the regulation of mitochondrial number and size via fusion and fission processes, ultimately offering a fascinating and perplexing glimpse into the inner workings of these organelles. The balance between these two processes is particularly important, as they control and regulate their formation and distribution, thereby mediating normal cellular functions and disease processes [1,2].

Mitochondrial fusion is the union of two mitochondria in one larger mitochondrion (Figure 1). They unite their outer membranes and then fuse their inner membranes. Each sub-compartment (inner, outer membrane, and matrix) blends with its respective sub-compartment from the other mitochondrion. The main components involved in mitochondrial fusion are also members of the dynamin family of GTPases (mitofusin 1 and 2, MFN1/2) in the outer membrane, and optic atrophy 1 (OPA1) in the inner membrane. MFN1/2 facilitate external membrane fixation and subsequent fusion [3,4]. Thus, mitochondrial fusion is considered beneficial as it is associated with an increase in mitochondrial function and ATP production.

Mitochondrial fission consists of the division of one mitochondrion into two separate organelles. The primary mediator of mitochondrial fission is dynamin-related protein 1 (DRP1, encoded by DNM1L in humans), a GTPase that is recruited to the mitochondrial outer membrane from the cytosol by binding to mitochondrial fission factor adapter proteins as well as 49 and 51 kDa mitochondrial dynamics proteins (MiD49/MiD51). After recruitment into the mitochondria, DRP1 performs fission by self-polymerization around the mitochondrial outer membrane, where it contracts the organelle in a process via GTP hydrolysis. Once assembled, DRP1 is sufficient to perform membrane constriction and cutting [3,5]. In contrast to mitochondrial fusion, mitochondrial fission appears to be harmful as it is associated with decreased mitochondrial functions and increased reactive oxygen species (ROS) production [6,7]. Damaged mitochondria produce more superoxide anion [8] and hydrogen peroxide [9], which potentiate mitochondrial ROS propagation [10].

Mitochondrial morphology can vary dramatically between cells and tissues. Deregulation of mitochondrial fission and fusion results in a fragmented network characterized by many small round-shaped mitochondria or hyperfused mitochondria with elongated and highly connected shapes. These balanced dynamic transitions are necessary to ensure normal mitochondrial functions and respond to cellular needs, adapting the network to nutrient availability and cell metabolic state [11].

Our review highlights the relationship between mitochondrial dynamics and the circadian cycle and the alterations resulting from the action of hypoxia and nicotine that lead to pulmonary hypertension (PH).

## 2. Role of Mitochondrial Fission in Pulmonary Hypertension

Mitochondrial fusion allows an exchange of material between the mitochondria, while fission allows for intact mitochondria to separate. Nutrient deprivation modulates mitochondrial morphology, thus stimulating mitochondrial fusion [12], while nutrient overload is often associated with mitochondrial fission [13]. Mitochondrial fusion contributes to the formation of an interconnected mitochondrial matrix, whereas mitochondrial fission leads to the formation of smaller mitochondria that are connected to each other or the endoplasmic reticulum. Several factors contribute to the regulation of mitochondrial fission and fusion, including cell-cycle kinases [14,15], phosphatases [16], cellular redox enzymes, intracellular calcium levels, calcium-dependent kinases [17], and metabolism. Mitochondrial dynamics are strongly linked to cell proliferation, apoptosis, and mitochondrial quality [18]. Both fission and fusion are also associated with various diseases including PH [19].

PH is defined as a mean pulmonary artery pressure (mPAP) ≥ 25 mmHg at rest [20,21]. The World Health Organization (WHO) has classified PH into the following five groups based on their origin, prognosis, and management: Group 1 mainly includes idiopathic or family pulmonary arterial hypertension; Group 2 merely occurs due to left heart disease; Group 3 is due to chronic lung disease (e.g., chronic obstructive pulmonary disease, interstitial lung disease, or overlap syndromes) or conditions that cause hypoxia (e.g., obstructive sleep apnea, alveolar hypoventilation disorders); Group 4 is called chronic thromboembolic pulmonary hypertension (CTEPH). CTEPH can occur when the body is not able to dissolve a blood clot in the lungs. This can lead to scar tissue in the blood vessels of the lungs, which blocks normal blood flow and makes the right side of the heart work harder; and Group 5 is a heterogeneous group of diseases that encompass PH secondary to multifactorial mechanisms, including sickle cell disease and sarcoidosis. Among these, PH attributable to left heart and lung diseases are the second most prevalent subtype with an estimated prevalence of up to 50% [22].

The enigmatic pathogenesis linking the mitochondrial dynamics to PH remains a challenge worldwide. Recent investigations have suggested the putative involvement of high-mobility group box-1 (HMGB1) and aberrant mitochondrial fission, resulting from the unwarranted activation of the DRP1, in patients with PH. Nonetheless, it remains a mystery whether DRP1-driven mitochondrial fission or its downstream targets provokes the HMGB1-induced migration and proliferation of PASMC in PH [23]. Further research to address this question is needed to better elucidate the mechanistic underpinnings of PH.

Changes in mitochondrial morphology have also been implicated in mitochondrial Ca^2+^ regulation and ROS production. ROS generation leads to inflammation and autoimmunity, which are important factors in the etiology of PH [24]. Mitochondrial morphology acts by regulating mitochondrial Ca^2+^ uptake to impact cellular Ca^2+^ homeostasis. Increasing fusion due to DRP1 knockdown increases the mitochondrial Ca^2+^ retention capacity. On the other hand, increasing fission because of MFN2 knockdown strongly reduces mitochondrial Ca^2+^ [25]. Altered mitochondrial dynamics play a direct role in pulmonary vascular dysfunction in PH through various switches such as increased oxidative stress and inflammation [19,26,27].

In PH, acquired mitochondrial abnormalities, including epigenetic silencing of superoxide dismutase (SOD2), disrupt oxygen sensing creating a pseudo-hypoxic environment characterized by normoxic activation of hypoxia-inducible factor-1α (HIF-1α). In addition, altered mitochondrial dynamics result in mitochondrial fragmentation. The molecular basis of this structural change includes the upregulation and activation of fission mediators, notably DRP-1, and the downregulation of MFN2. These pathogenic mitochondrial abnormalities offer new therapeutic targets. Inhibition of mitotic fission or enhancement of fusion in PH PASMC slows cell proliferation, causes cell cycle arrest, and induces apoptosis. DRP-1 inhibition or MFN2 gene therapy can regress PH [28].

## 3. The Circadian Clock and Its Regulation of Mitochondrial Dynamics Are Critical to Pulmonary Vascular Remodeling and PH

Although the role of the circadian clock has been well described in the progression of chronic airway diseases after exposure to nicotine, interruption of the circadian cycle has not been associated with pulmonary vascular remodeling, resulting in PH [29]. Unfortunately, little is known about the circadian variability that directly contributes to PH, despite the interest in intact lung effects [30]. Energy, oxidative, and anti-oxidative metabolism are coordinated by the circadian clock. Alterations in these mechanisms can impair metabolic homeostasis [31].

The pivotal role of mitochondrial oxidative stress leading to an alteration in pulmonary vascular pathophysiology has also been recently investigated. It has been demonstrated in detail that circadian clock plays a significant role in the regulation of tissue physiology through oxidative-reductive mechanisms [32,33,34,35,36]. For instance, the NAD^+^-dependent histone deacetylase silent information regulator (SIRT3)-dependent circadian modulation has been strongly linked to mitochondrial oxidative metabolism. Pekovic et al. have developed a novel model to establish a link between pulmonary pathology and mitochondrial circadian clock; according to their data, the circadian clock plays a major role in modifying the pulmonary fibrotic response via modulation of the NRF2 transcription factor [37]. Therefore, investigating the role of circadian modulation of the NRF2 can help in establishing new therapeutic targets for the treatment of pulmonary diseases such as pulmonary fibrosis.

Circadian clocks confer a cell, organ, and/or organism to prepare for a stimulus or stress before its onset, thus, facilitating a temporally appropriate response. The crucial components of the circadian clock include two transcription factors, CLOCK (circadian locomotor output cycles kaput) and BMAL1 (brain and muscle ARNT-like 1). By heterodimerization, these two molecules bind to E-boxes in the promoter of target genes, invariably leading to induction. Target genes include negative feedback components such as period isoforms [38], cryptochrome (CRY1/2), and Rev-erb (REV/ERBα/β); the PER/CRY heterodimer complexes repress the transcriptional activity of CLOCK/BMAL1 through physical interactions, while REV-ERBα/β represses BMAL1 at the transcriptional level [39,40].

Several studies propose that mitochondrial functions and morphological changes are dependent on a viable circadian clock. Rev-erbα modulates the oxidative capacity of skeletal muscle by regulating mitochondrial biogenesis and autophagy [41]. An altered mitochondrial gene expression and increased oxidative stress were discovered in mice with clock gene mutations [42]. Furthermore, mitochondrial fission and fusion exhibit a significant change with the light/dark cycle. Disrupting the molecular clock in animal models leads to abrogated mitochondrial rhythmicity and altered respiration. However, further experiments are needed for the better understanding of how mitochondrial morphology and function change throughout the day [43].

Although the role of mitochondria in ROS production and cellular redox regulation are well known, little is known about the role of circadian genes, including Rev-erbα, in modulating these two mitochondrial cellular responses. Even so, it is known that the central circadian gene, Rev-erbα, regulates mitochondrial energy production and enhances cellular antioxidant defense, thereby protecting cells against oxidative stress. This energetic advantage helps maintaining cell homeostasis under adverse conditions to promote cell survival and growth [8].

An intriguing finding is the discovery of a secondary feedback loop that links mitochondrial H_2_O_2_ production to intracellular signaling [43]. The nuclear receptors Rev-Erb are involved in many physiological processes, including circadian rhythm, metabolism regulation, and inflammatory functions, including bile acid metabolism, lipid metabolism, and inflammatory cytokine production [44,45]. Woldt et al. have demonstrated that the genetic deletion of Rev-erbα in mice results in increased mitochondrial fission, impaired oxidative function, and decreased mitochondrial content; moreover, the number of mitochondria and respiratory capacity are improved after treatment with the vector expressing Rev-Erbα; in contrast, Rev-erbα overexpression can improve ex vivo mitochondrial respiration [41]. Woldt have further demonstrated that mitochondrial DNA content is lower by 40% in skeletal muscle from Rev-erbα^−/−^ mice compared to littermates, suggesting a regulatory role of Rev-erbα in skeletal muscle mitochondrial content.

Consequently, the expression of genes encoding mitochondrial electron transport chain subunits, such as a complex I subunit (NADH dehydrogenase 1) and complex IV subunits (cytochrome c oxidase-1 and 2), are lower in the quadriceps of Rev-erbα^−/−^ mice [41], suggesting a likely reduction in gene expression of the important complex III subunit Rieske iron-sulfur protein [46]. Rev-erbα plays a key role in regulating swings in BMAL1 expression via directly binding to the BMAL1 promoter and suppressing its expression at certain times of day when Rev-erbα expression levels are high [47]. The CLOCK-BMAL1 complex induces transcription of Rev-erbα and Rev-erbβ, which subsequently compete with retinoic acid receptor-related orphan receptors (ROR) to inhibit transcription of CLOCK and BMAL1 [43].

Mitochondrial dynamics and biogenesis are transcriptional targets of the circadian regulator Bmal1 in the mouse liver and exhibit a metabolic rhythm coordinated with diurnal bioenergetic demands. Loss of Bmal1 function causes bloated mitochondria, unable to adapt to different nutritional conditions, accompanied by decreased respiration and heightened oxidative stress [48]. Newly formed mitochondria fuse to form a tubular network. Mitophagy is preceded by mitochondrial fission to form fragmented mitochondria that can be taken up by an autophagosome. Both fusion and fission are influenced by CLOCK-BMAL1. Several regulatory proteins regulate the processes of fusion (OPA1, MFN1/2), fission (FIS1, DRP1), and mitophagy (PINK1, BNIP1, PARKIN) [43]. Indeed, to maintain mitochondrial homeostasis, mitochondria continually change their dynamics, increasing or decreasing fission and fusion processes. The CLOCK-BMAL1 complex stimulates mitochondrial biogenesis and mitophagy through the activation of SIRT1.

Mitochondrial biogenesis is regulated by the transcription factor PGC1A [43]. In addition to SIRT1, SIRT3 is the primary deacetylase in mitochondria and is involved in both metabolic homeostasis and stress response. While the protein level of SIRT3 is constant, its activity is regulated in a diurnal manner because of oscillating levels of coenzyme NAD in the heart. This subsequently affects key mitochondrial functions, including metabolism and oxidation reduction. Circadian regulation of NAD allows for coordinated catabolism and ROS scavenging, being essential for myocardial homeostasis [49]. SIRT3^−/−^ mice show a reduction of more than 50% in basal ATP levels, fatty acids oxidation, and glucose oxidation. However, an increase in glycolysis is detected in cardiac cells. These changes are accompanied by the hyperacetylation of many mitochondrial proteins that are involved in fatty acid transport and oxidation, electron transport chain subunits, and tricarboxylic acid cycle [50].

REV-ERBα is known to suppress CCL2 chemokine expression by cells of the monocyte/macrophage lineage [29], which are known to be both necessary and sufficient for the development of hypoxia-mediated PH [51] through the native CCR2 receptor [52]. Likewise, REV-ERBα directly binds to the IL6 promoter region and indirectly affects NF-κB binding motif to inhibit IL-6 signaling [53], which is considered to have a protective effect in patients with PH [54]. Expression of REV-ERBα by myeloid cells also protects against the development of atherosclerotic plaque formation, with deficiency of the protein resulting in more severe myocardial infarction [55].

## 4. Hypoxia Decreases the Amplitudes of Circadian Oscillations, Contributing to PH

Hypoxia is a crucial factor in altering mitochondrial dynamics. It has been reported that hypoxia induces mitochondrial fragmentation or a smaller mitochondrial size [56]. Furthermore, the mitochondrial number and area are increased significantly, while the mitochondrial length/width ratio is significantly reduced in pulmonary vascular tissue in hypoxia-induced PH.

Recent discoveries reveal distinctive structural and functional deviations of mitochondria within PASMC in both human and animal PH. These untypicalities entail mitochondrial fragmentation and aerobic glycolysis, which collectively contribute to the profound phenotypic characteristic of PH [57,58]. The apoptosis-resistant, proliferative phenotype of PASMCs and the bioenergetic impairment of right ventricle myocytes are both mediated by mitochondrial fragmentation and aerobic glycolysis, which is similar to those in cancer patients [59].

As mentioned above, mitochondria form a tubular network whose morphology is linked to homeostasis in fission and fusion events. As fusion and fission are processes that can be altered by different metabolic conditions, hypoxia has an important role in their regulation. The lack of oxygen induces the cell to consume more glucose to produce ATP through anaerobic glycolysis. Under these conditions, mitochondria may not receive sufficient substrates, such as acetyl-CoA and O_2_, inducing major changes in structure, function, and dynamics [60]. Studies further show that defective mitochondrial fusion leads to mitochondrial depolarization, loss of mtDNA, altered respiration rate, impaired distribution of mitochondria [61], and inhibited mitochondrial motility [62]. Similarly, exposure of cortical neurons to mild hypoxic conditions for several hours significantly altered mitochondrial morphology, decreased mitochondrial size, and reduced mean mitochondrial velocity [63].

Mitochondria-dependent ROS, which serves as a primary signaling molecule to mediate the development of PH, has been linked to the circadian clock. Accumulating evidence indicates that mitochondrial ROS production and elimination show diurnal fluctuations like mitochondrial oxidative phosphorylation [43]. Mitochondria are an oxygen sensor responsible for controlling ROS production and cytosolic redox state in PASMCs, which may regulate ion-channels, receptors, and kinases, thereby playing an important roles in altering pulmonary vascular responses following hypoxia [64]. Intriguingly, hypoxia-induced mitochondrial ROS occur earlier and stronger in PASMCs; as a result, mitochondrial ROS can activate NADPH oxidase (NOX) by increasing protein kinase C-epsilon (PKCε) to induce more ROS formation, a mechanism known as ROS-induced ROS generation (RIRG) [46].

Acquired mitochondrial defects, such as epigenetic suppression of superoxide dismutase, alter oxygen sensing in PH and result in a pseudo-hypoxic milieu, which is marked by normoxic activation of hypoxia-inducible factor-1 (HIF-1) and causes a metabolic shift to aerobic glycolysis (the Warburg phenomenon) due to pyruvate dehydrogenase kinase inhibition. Furthermore, abnormal mitochondrial dynamics lead to mitochondrial fragmentation. Upregulation and activation of fission mediators, particularly DRP1, and downregulation of fusion mediators, particularly MFN2, are at the molecular basis of this structural shift. These pathogenic mitochondrial defects present novel therapeutic opportunities. In PH PASMC, inhibiting mitotic fission or increasing fusion inhibits cell growth, causes cell cycle arrest, and promotes apoptosis.

Persistent oxidative stress due to circadian regulation of mitochondrial determinants has been strongly linked to pulmonary endothelial dysfunction leading to PH. Data suggests that vascular endothelium is one of the major sources of ROS generation in human blood vessels [65].

Studies in rats and humans show that the most common effect of prolonged hypoxia decreased the amplitudes of daily oscillations, regardless of arousal state or activity level. Furthermore, hypoxia modifies very important variables such as body temperature and metabolism, expecting the daily rhythms of many other functions to be disturbed by hypoxia [66]. Hypoxia alters thermoregulation; indeed, in rats, one of the consequences is a change in circadian variations in body temperature. These hypoxic effects also apply to acute human exposure [67]. Circadian misalignment can also occur between the clocks of different peripheral tissues following exposure to hypoxia [68].

Bosco et al. [67] have showed that the circadian oscillation amplitudes of tympanic calf skin temperatures are reduced during hypoxia, averaging 61%, 80%, and 50% of the normoxic amplitude, respectively. Oxygen consumption and pulmonary ventilation have a circadian pattern under normoxic conditions, they no longer show significant oscillations during hypoxia, while the heart rate and diastolic pressure have opposite effects [67]. Furthermore, hypoxia increases the expression of the circadian genes PER1 and CLOCK in the mouse brain, modulating the circadian clock at the molecular level, probably via affecting HIF-1 signaling [38]. Through the action of tobacco smoke on lung parenchyma cell rhythms, the pulmonary vasculature would also have a specific change at the tissue level at the clock output to further response to hypoxia [29]. Noticeably, circadian transcriptomic changes in hypoxic lung immune cells are very similar to those in patients with PH [69].

New drugs that have emerged targeting mitochondrial dynamics controlled by the clock have brought hope for therapeutic improvements. The first drug based on the circadian clock was melatonin, a hormone secreted by the pineal gland [70]. Exogenous melatonin supplementation increased myocardial ROR levels through the reduction of dysfunctional mitochondria, ER stress, myocardial apoptosis, autophagy dysfunction, and oxidative stress damage [70,71].

Furthermore, in recent years new therapeutic alternatives have been presented, such as nobiletin, CRY activator (KL001), Rev-ERB-α/β agonist (especially SR9011/SR9009), among other small molecule chemical enhancers targeted at the circadian system. These chemical compounds restored cardiac circadian rhythms and oscillatory patterns of metabolic gene expression, resulting in phenotypic improvements in insulin resistance, lipotoxicity, oxidative stress, and dysfunctional mitochondrial dynamics [70,72,73]. Therefore, interventions involving the circadian clock and mitochondrial dynamics are promising therapeutic approaches.

## 5. Mitochondrial Dynamic Dysfunction in Hypoxia Contribute to Group 3 PH

Changes in mitochondrial dynamics are correlated with decreased mitochondrial respiratory enzyme activity and ATP abundance in hypoxic PH. Moreover, PASMCs from PA present dysmorphic mitochondria with reduced respiratory chain coupling, inefficient use of oxygen, and increased glycolysis [56]. These pathological changes were significantly attenuated in knockdown HIF-1α cells. The HIF-1α knockdown also significantly increases the mitochondrial length/width ratio, reducing the number of mitochondria, and the mitochondrial area [56].

After hypoxia stimulation, lung tissues and PASMCs present reduced mitochondrial respiratory complex IV and ATP, increased ROS production, and down-regulated mtDNA content, indicating hypoxia-induced deficiencies in energy metabolism and enzymatic activity of the respiratory complex [56]. Hypoxia also promotes mitochondrial retention in the perinuclear area, partial microtubule dissociation, anomalous fusion activity, and suppression of general fusion activity, leading to mitochondrial shortening. A key trigger for this response is the suppression of mitochondrial ATP production and the depletion of cellular ATP [74].

The expression of the fusion proteins MFN1/2 is significantly reduced in pulmonary artery tissues from rats with PH. In addition, MFN1/2 mRNA levels are significantly decreased in PASMCs after exposure to hypoxia compared to normoxia. These results suggest that mitochondrial fusion proteins MFN1/2 are involved in pulmonary vascular remodeling and PH [56]. The expression levels of the DRP1 fission protein also show a significant increase in PASMCs following hypoxia, both in vivo and in vitro. This suggests that the mitochondrial fission protein DRP1 is involved in pulmonary vascular remodeling. HIF-1α mediates the hypoxic activation of DRP1 to result in the induction of the mitochondrial pathway of proliferation and apoptosis in PASMCs via modulation of protein cell nuclear antigen (PCNA) and caspase-3 expression [56].

Nox4 is a major subunit of NADPH oxidase. Its expression is increased in murine models of hypoxia-induced PH in the pulmonary vasculature of patients with PH [75]. Hypoxia increases Nox4, which promotes the increase of mitochondrial hydrogen peroxide, promoting PH [76]. Nox4 silencing by siRNA causes reduction of ROS levels under normoxic and hypoxic conditions and suppresses the hypoxia-induced significant ROS increase in pulmonary adventitial artery fibroblasts [77].

Studies have also shown that Nox4 is found within mitochondria [78,79]. Nox4 modulates the activity of enzyme complexes within the electron transport chain (ETC) [80], in addition to its interaction with complex I, inhibiting its activity [9]. As hydrogen peroxide (H_2_O_2_) is the main product of Nox4 activity, the presence of active Nox4 in mitochondria can be expected to increase mitochondrial H_2_O_2_ levels. Koziel et al., demonstrated a significant decrease in the concentration of H_2_O_2_ in the mitochondria of Nox4-knockout cells [81]. As another subunit of NADPH oxidase, an important finding regarding Nox4 is its role in mitochondrial morphology, as in Nox4-knockout cells mitochondria revealed a characteristic of highly separated, non-interconnected networks [81]. Furthermore, the increase in mitochondrial Nox4 expression induces vascular SMC-mediated structural remodeling of the vascular wall due to the increase in ROS [82].

PKCε is a cytosolic protein that can be translocated to mitochondria in certain situations, even though little is known about this mechanism [83]. PKC-ε also appears to play an important role in mitochondrial morphology. An interesting study established that PKC-ε activation in renal proximal tubular cells induces mitochondrial dysfunction and fragmentation, energy deficit, ROS generation and cell death [84]. Rieske iron-sulfur protein (RISP) is a primary key factor in the generation of ROS originated from mitochondria. Our studies and others reveal that RISP is an important primary molecule for initiating hypoxia-induced [ROS]_mito_ generation in PASMCs, cardiac myocytes, and neuronal cells [85]. We further demonstrate that [ROS]_mito_ can subsequently activate cytosolic PKCε and then cell membrane NOX to induce further ROS generation; this ROS-induced ROS production ultimately cause massive increases in [ROS]_i_ in PASMCs [46,86].

RISP-mediated increase of intracellular ROS may subsequently inhibit voltage-gated potassium (Kv) channels, and activates transient receptor potential (TRP) channels, ryanodine receptors (RyRs) (especially RyR2) as well as inosol 1,4,5-trisphosphate receptors (IP3Rs) to evoke a large increase in [Ca^2+^]_i_, leading to numerous cellular responses. It is worth pointing out that RyRs may mediate the hypoxic inhibition of KV channels, activation of TRP channels, and amplify IP3R-dependent Ca^2+^ release [87,88]. Our very recent investigations discover that RyR2-mediated Ca^2+^ release from the sarcoplasmic reticulum can further promote mitochondrial ROS generation [46].

The pathophysiologic relevance and therapeutic implications of defective mitochondrial fusion and excessive fission in PH have been widely studied. PASMCs play a critical and central role in the development, progression, and advancement of PH. The mitochondria in PASMCs play a critical role as vascular sensors of oxygen and thus may respond to a slight increment or decrement of oxygen tension in the pulmonary artery by activating hypoxic pulmonary vasoconstriction (HPV) [89,90]. HPV is the mechanism employed by the lungs to rectify any ventilation-perfusion mismatch [64]. A decline in oxygen levels in the pulmonary vasculature gives rise to vasoconstriction and shunts the blood-flow towards promoting better perfusion in well-ventilated lung areas. HPV is also important in optimizing partial pressure of CO_2_ in certain lung conditions including atelectasis and pneumonia [91]. The PASMCs inside these arterial segments have mitochondria that behave as though they have been exposed to continuous hypoxia, and the pathophysiology of PH is centered in these same resistant arteries. In particular, the mitochondria in PH PASMCs exhibit poor metabolism because of transcriptionally mediated inhibition of mitochondrial pyruvate dehydrogenase and are fragmented as a result of an imbalance between mitochondrial fission and fusion.

The hypothesis known as the “redox hypothesis” proposes that the ROS production from the electron transport chain (ETC) complexes I and III in the mitochondria is altered in direct correlation to the level of alveolar PO_2_, thereby initiating HPV in the human body [92]. The disruption of electron flow and reduction in levels of the diffusible second messenger H_2_O_2_ serve as markers of acute hypoxia [93]. The consensus among experts is that the mitochondria act as a primary oxygen sensor in the body, monitoring the levels of ROS in the system as an indicator of hypoxia. However, some dissenting opinions suggest that there may be a paradoxical increase in ROS levels during hypoxia [94].

A set of scientific data reveal that the synthesis of diffusible redox mediators, including radicals and peroxides, is diminished specifically in the resistance PASMC (as opposed to the conduit artery PASMC) during physiologic hypoxia (as opposed to anoxia). As such, hypoxia may lead to a decrease in the generation of ROS, which, in turn, causes inhibition of Kv channels, membrane depolarization, activation of voltage-gated L-type calcium (Cav1) channels, depolarizing the PASMCs. Vasoconstriction begins because of the calcium entry [64,92,95].

The occurrence of HPV in resistance PAs can be attributed to the distinctive capacity of PASMC mitochondria to regulate their ROS production [96]. This phenomenon is not observed in other arteries such as the renal arteries, where the production of ROS in response to changes in PO_2_ is not significantly altered, and hypoxia results in arterial dilation instead [96]. The precise role of mitochondrial dynamics in HPV is yet to be fully comprehended. However, mitochondrial fission is an obligatory preliminary step in the mechanisms that precede changes in mitochondrial ETC function and ROS signaling in the ductus arteriosus [97]. Although HPV can cause acute PH and contribute to diseases like high altitude pulmonary edema in genetically predisposed people, it is imperative to understand that the underlying mechanism that leads to PH is ROS generation [98]. Hypoxia suppresses the mitochondrial pathways of ROS generation that underlie the oxygen-sensing function of mitochondria. More specifically, chronic hypoxia activates HIF-1α which in turn decreases mitochondrial H_2_O_2_ production and minimizes PO_2_-sensitive ROS generation. This in turn decreases HPV [99].

PH related to chronic hypoxia may also primarily result from the remodeling of pulmonary vessels (ex. medial hypertrophy of small pulmonary arteries (<200 μm) in addition to HPV [64]. This vasculopathy in PH is caused by abnormalities in redox signaling (activated HIF-1 and decreased SOD2), oxidative metabolism (increased PDK and inhibited PDH), mitochondrial dynamics (increased dynamin related protein 1, DRP1, and reduced MFN2), and effector targets (altered expression of O_2_-sensing K_v_ channels). Warburg hypothesis proposes that this impaired O_2_-sensing impairment contributes to the underlying pathology in PH in a similar pattern like cancer relies on glycolysis despite availability of oxygen availability for oxidative metabolism. This theory suggests that both PH and cancer are reliant on the failure of oxygen-sensing due to the alteration of mitochondrial redox functions. This, in turn, manifests as further impairment of oxygen-sensing which in the longer term gives rise to the Warburg phenomenon [100].

The role of the Cyclin B and CDK1 complex has been investigated to play a role in PH. Cyclin B-dependent CDK1 initiates a cycle of mitosis by phosphorylating DRP1 at serine 616, therefore activating mitochondrial fission [101]. On the other hand, inhibition of mitotic fission arrests the cell-cycle at G2/M transition, promoting cell death [102]. However, whether hypoxic fragmentation of mitochondria leads to vasoconstriction in PH is still largely unknown. In ductus arteriosus, a change in PO_2_ results in rapid (<60 s) mitochondrial fission, ultimately resulting in ROS production, inflammation, and vasoconstriction [97].

Von-Hippel Lindau disease (VHL), a genetic disease discovered in the Chuvash region of Russia, is a striking example of the impairment of oxygen-sensing in mitochondria as the underlying pathophysiology of pulmonary hypertension. A loss-of-function mutation in the VHL factor gives rise to pulmonary hypertension due to normoxic activation of HIF-1α [103,104]. The spectrum of symptoms in VHL comprises polycythemia and PH despite normal oxygen levels. This phenomenon demonstrates that impaired oxygen sensing (and the resulting normoxic activation of HIF-1α and HIF-2α) is sufficient to cause PH [103].

Mitochondrial redox signaling mechanisms in PH create a transcriptional and proteomic fingerprint like that observed in sustained hypoxia. These abnormalities are seen redundantly despite high-oxygen conditions, such as in cell-culture, which provide a state of pseudohypoxia.

PH-associated abnormalities in the pulmonary vascular oxygen-sensing pathway include persistent activation of HIF-1α during normoxic conditions [57] and a transcriptional activation of enzyme PDK in pulmonary arteries [57,96] and RV [105,106]. This causes a rapid shift from oxidative metabolism to aerobic glycolysis, and in turn, impaired mitochondrial fusion, and impaired fission, which results in the fragmentation of the PASMC’s mitochondrial network [102,107].

## 6. Nicotine Induces Mitochondrial Fission through Mitofusin Degradation, Leading to PH

Nicotine is the main active component of tobacco smoke. Through interactions with nicotinic acetylcholine receptors (nAChRs) in the central nervous system, nicotine modulates several genes and cellular pathways [108]. However, nicotine may affect the electron transport ETC of mitochondria independently of nAChRs [109]. Although many different nAChR receptors have been studied, the α7nAChR is the key subtype receptor involved in nAChR-mediated immune regulation [110,111]. In the past, α7 nAChRs were believed to be present only on the plasma membrane; however, the discovery of α7nAChRs in intracellular organelles such as mitochondria contradicts this dogma [112]. Mitochondrial α7nAChRs help to regulate mitochondrial pore formation as well as control the mitochondria-induced apoptotic process [113]. Previous studies showed that nicotine induces mitochondrial fission through mitofusin degradation [6,114] and mitochondrial dynamic imbalance and apoptosis through mitophagy impairment, by repressing cathepsin L activity and activating ROS-mediated p38/JNK pathways [6].

Controversial data have been generated regarding the action of nicotine on mitochondria depending on the target tissue. Cormier et al. [109] have demonstrated that nicotine is able to inhibit the generation of ROS by 15.7% in rat brain tissue, which may occur due to its competition with NADH in complex I of the cerebral mitochondrial ETC to decrease the generation of ROS.

On the other hand, nicotine is associated with an increased risk of cardiovascular disease, promoting vascular endothelial dysfunction, increased oxidative stress, inflammation and apoptosis, which may contribute to PH [115]. Studies show that exposure to nicotine increases the production of mitochondrial-derived ROS in cardiomyocytes. In addition, nicotine significantly promotes DRP1-mediated mitochondrial fission and suppressed mitofusin-mediated fusion in cardiomyocytes [6]. Several studies have shown that nicotine administration increases NADPH oxidase and mitochondria-driven ROS production in cardiomyocytes [116]. Maternal exposure to nicotine (1 mg nicotine/kg body weight/day, subcutaneously) results in the swelling of the mitochondria of the alveolar septa in offspring from 1 to 21 days [117].

Nicotine can directly or indirectly trigger various intracellular signaling pathways. Moreover, nicotine has an impact on the variation of cytosolic calcium concentration, which may influence mitochondrial calcium homeostasis. Ca^2+^ homeostasis is fundamental to the function of mitochondria and is related to the generation of ATP, ROS, and mitochondrial dynamics [118,119].

Nicotine may increase intracellular calcium entry via nAChRs, thus reducing the mitochondria membrane potential and inducing mitochondrial translocation of E3 ubiquitin ligases; this increases the proteasomal degradation of MFN1/2. These effects can be blocked by treatment with mecamylamine, a nonselective nAChR antagonist. These data suggest that nicotine degrades MFN and thus induces mitochondrial dysfunction and cell growth inhibition in a nAChR-dependent manner [114]. 

Calcium entry via α7nAChR, via Phosphoinositide 3-kinases/Akt-dependent classical transient receptor potential 6 signaling pathway plays an important role in the physiological regulation of airway smooth muscle cell proliferation, representing an important target for augmenting airway remodeling [120]. Nicotine elevates [Ca^2+^]_I_ in ASMCs through α7nAChR-mediated signals pathways, and highlight the possibility that α7nAChR can be considered as a potential target for the treatment of airway remodeling and associated diseases [121]. It seems that modulation of mitochondrial dynamics by nicotine regulates Ca^2+^ efflux from the mitochondria through the Na^+^/Ca^2+^ exchanger to continuously fill the ER. Thus, repeated nAChR activation can enhance calcium release from mitochondria [118].

PH is an idiopathic cardiovascular disease characterized by vascular cell proliferation, apoptosis-resistance, inflammation, thrombosis, and vasoconstriction, which block the small pulmonary arteries [122]. Chronic nicotine use promotes the development of vascular dysfunction by decreasing the formation and bioavailability of nitric oxide (NO) [123,124,125]. Additionally, nicotine inhibits eNOS, an enzyme that produces NO, reduces endothelium-dependent vasodilation, and promotes leukocyte adhesion in the endothelium, resulting in vascular endothelial dysfunctions (VED) and subsequently atherosclerosis [125,126,127]. Additionally, nicotine causes hypertension by promoting catecholamine release from chromaffin cells [128]. By generating coronary artery spasm and raising the demand for myocardial oxygen, nicotine exposure from cigarette smoking (CS) has been recognized to play a pathogenic role in the formation of ischemic myocardium [19]. Additionally, nicotine triggers apoptosis in cardiomyocytes [6].

The link between nicotine use and mitochondrial influences has been extensively studied. Nicotine inhibits the oxidative phosphorylation (OXPHOS) system. Cormier et al. have showed that nicotine reduces the oxygen consumption of isolated rat brain mitochondria in a concentration-dependent manner as a low as a concentration of 1 × 10^12^ M, with a maximum inhibitory effect at 1 × 10^7^ M [109]. This effect establishes its direct influence on mitochondrial ETC complexes.

This aspect of nicotine’s effect on mitochondrial oxygen consumption establishes its direct influences on mitochondrial respiratory chain complexes. Smokers are 2–4 times more likely than non-smokers to suffer cardiovascular and pulmonary diseases (CVPD), making cigarette use the single most significant risk factor. Nicotine has been associated with an increased risk of chronic obstructive pulmonary disease (COPD), cardiovascular disease, promoting pulmonary hypertension, vascular endothelial dysfunction, increased oxidative stress, inflammation, and apoptosis in most industrialized countries [129,130,131]. Vascular endothelial dysfunctions (VED) have been considered hallmarks of various cardiovascular disorders [132,133].

## 7. Nicotine Alters Mitochondrial Dynamics, Increasing COPD

COPD is a debilitating inflammatory lung disease characterized by irreversible deterioration of the airway wall, airflow limitation, chronic bronchitis, emphysema, and airway remodeling [134,135]. Decreased lung function in COPD is associated with an increase in bronchial smooth muscle, which is probably the most important abnormality responsible for airway narrowing in response to bronchoconstriction stimuli [121]. Cigarette smoke is considered the main risk factor for the development of COPD; however, genetic factors, host responses, and infection also play an important role [135].

Furthermore, cigarette smoke extract (CSE) induces mitophagy in airway epithelial cells through oxidative stress. CSE negatively regulates the expression of type I collagen and α-SMA, but positively regulates fibronectin. CSE decreases PGC-1α, MTCO2, and MTCO4, but increases beclin-1, p62, and LC3. CSE positively regulates mitophagy and lysosome activity via phosphorylation of ERK1/2. In vitro, cigarette smoke induces deterioration of ASMCs, which may explain tissue loss and structural remodeling in COPD bronchi [134].

As already mentioned, increased ROS has been implicated in the pathophysiology of COPD. COPD macrophages exhibit defective phagocytosis, which is associated with altered mitochondrial function and the inability to regulate mitoROS production [136]. Furthermore, evidence highlights cellular iron accumulation in the lung, due to changes in mitochondrial dynamics, as a key contributing factor in the development and pathogenesis of COPD [135].

COPD is a destructive inflammatory disease and genes expressed in the lung are crucial to its pathophysiology. Cigarette smoking causes hypomethylation of the aryl hydrocarbon receptor repressor (AHRR) gene, which regulates detoxification and responses to oxidative stress [137]. Increased airway epithelial AHRR expression may lead to cigarette smoke-induced mitochondrial dysfunction and apoptosis [137].

Another interesting fact that has been recently reported is sex-specific gene expression in individuals with COPD. Analysis of differentially expressed genes (DEGs) in women with COPD revealed many genes with strong differential expression that either show no differential expression in men, or show a reverse pattern [138].

Understanding the molecular mechanisms underlying changes in mitochondrial dynamics may reveal potential new avenues of investigation and therapeutic targets to aid in the treatment of COPD.

## 8. Nicotine Exacerbate Circadian Disturbances in Lung Function, Contributing to Vascular Remodeling and PH

Several studies suggest that nicotine effects may be influenced by the circadian system [139]. Administration of nicotine in rodents, a dose comparable to that produced by a single CS, strongly alters neuronal activity in the suprachiasmatic nucleus (SCN), the locus of the circadian clock in mammals [140]. Furthermore, Morley and Garner [141] have showed that nicotine causes an increase in locomotor activity in the light phase in rats, but not in the dark phase. Rev-erbα plays a crucial role in regulating CS-induced lung inflammation and injury. Rev-erbα cell-specific deletion exacerbated the pathogenesis of emphysema/COPD caused by CS. The observation from Rev-erbα/β mice demonstrated the potential anti-inflammatory role of Rev-erbα/β in lung epithelium cells [27].

Lung clock alteration by environmental agents and tobacco smoke can have repercussions in the pathophysiology of COPD and its exacerbations [142]. Exposure to nicotine contributes to the altered expression of circadian molecular clock genes in mouse lungs, which may have repercussions on lung cellular and biological functions [143].

Expression of Rev-erbα gene in lung tissues is upregulated in nicotine exposed mice, suggesting that nicotine exposure induces changes in pulmonary clock gene expression [143]. A study in the murine model has shown that nicotine can affect the transcriptional regulation of Rev-erbα, thereby disrupting lung circadian rhythmicity that may play an important role in nicotine-induced lung pathophysiology [144].

Patients with COPD have abnormal circadian rhythms in lung function. BMAL1 is acetylated and degraded in the lungs of mice exposed to CS and in patients with COPD. Exposure to CS altered clock gene expression and reduced locomotor activity, disrupting central and peripheral clocks and increasing lung inflammation, causing emphysema in mice [145]. BMAL1 plays a key role in lung epithelium, mediated by SIRT1-dependent BMAL1, in the regulation of nicotine-induced inflammatory and harmful lung responses. Targeted deletion of Bmal1 in lung epithelium increases inflammation in response to CS; However, this response is not attenuated by the selective SIRT1 activator SRT1720 in these mice [145].

Through the action of tobacco smoke on lung parenchyma cell rhythms, it stands to reason that the pulmonary vasculature would also have a specific change at the tissue level at the clock output in response to exposure to inhalants, temperature, hypoxia, etc. [29]. Incidentally, it has recently been shown that circadian transcriptomic changes in hypoxic lung immune cells are very similar to those in patients with PH [69].

## 9. Future Perspectives and Conclusions

Hypoxia and nicotine can directly or indirectly trigger several intracellular signaling pathways. As diagrammed in Figure 2, hypoxia or nicotine exposure can elevate RISP-mediated mitochondrial calcium and ROS production by increasing DRP1 and reducing MFN1/2 degradation, thereby leading to increased mitochondrial fission. Nicotine also increases ROS and then decreases mitochondrial fusion, inducing mitochondrial fragmentation, and an increase in undersized mitochondria, in addition to stimulating mitophagy. Important proteins in the fission process, such as DRP1, are present in greater quantities, while there is a reduction in MFN1/2 proteins, for fusion. This increase in mitochondrial calcium and ROS leads to increased calcium and ROS in the cytosol, leading to increased vasoconstriction and vascular remodeling, culminating in pulmonary hypertension. Furthermore, hypoxia can also alter the circadian cycle, increasing the expression of genes such as PER1 and CLOCK, decreasing the amplitude of daily oscillations, and altering body temperature. Mitochondrial functions and morphological changes depend on a viable circadian clock. Mitochondrial dynamics change according to the light/dark cycle. Alteration of this cycle leads to serious problems in mitochondrial homeostasis, leading to neurodegenerative diseases, such as Alzheimer’s disease. Mitochondria play a crucial role in several cellular processes, and maintaining their balance through circadian rhythms and in response to nicotine and hypoxia is vital for preventing oxidative stress and inflammation and understanding and treating pulmonary vascular dysfunction and hypertension.

Circadian biology has many important cellular functions in the pulmonary vasculature, including smooth muscle and endothelial cell mitochondrial metabolism proliferation and contraction, in response to nicotine and/or hypoxia. There is much evidence to suggest an important role for the correlation of circadian biology and mitochondrial dynamics in the initiation and progression of almost all lung diseases. An in-depth understanding of circadian roles during nicotine exposure and hypoxia is crucial to fully elucidate their underlying precise molecular mechanisms. These findings could also help develop new specific and effective therapeutic strategies for treating relevant PH and other lung diseases.

## Figures and Tables

**Figure 1 biomedicines-12-00053-f001:**
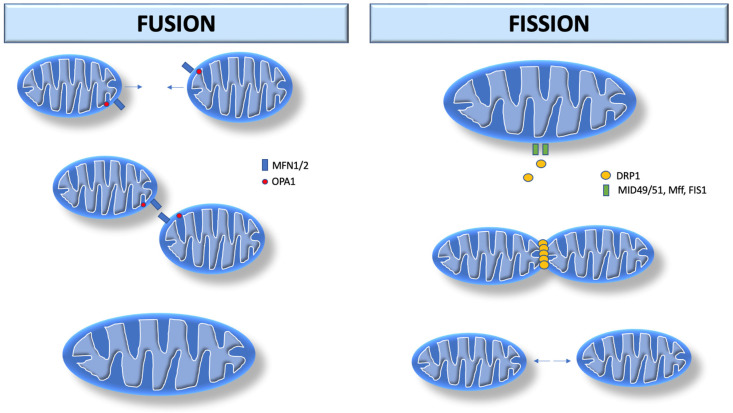
Mitochondrial dynamics. Mitochondrial fusion begins with MFN1/2 in the outer membrane, and OPA1 in the inner membrane. MFN1/2 facilitates external membrane fixation and subsequent fusion. Mitochondrial fission begins after recruitment of DRP1 by MiD49/MiD51. DRP1 performs fission by self-polymerization around the mitochondrial outer membrane, where it contracts the organelle in a process that uses GTP hydrolysis. MFN1/2: Mitofusin 1 and 2; OPA-1: optic atrophy 1. DRP1: dynamin-related protein 1; MiD49: mitochondrial dynamics protein 49 kDA; MiD51: mitochondrial dynamics protein 51 kDA; Mff: mitochondrial fission factor; FIS1: fission protein 1.

**Figure 2 biomedicines-12-00053-f002:**
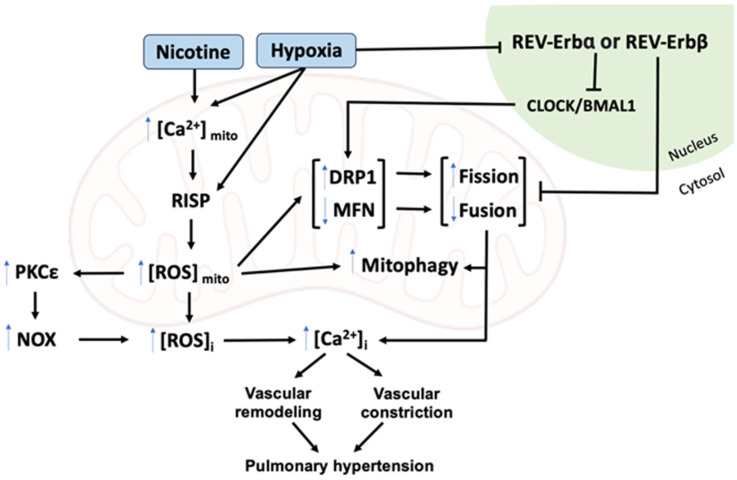
Schematic of signaling pathways for nicotine/hypoxia-mediated cellular responses. Hypoxia or nicotine exposure can elevate RISP-mediated mitochondrial calcium concentration and ROS production leading to increased mitochondrial fission and mitophagy. These mitochondrial responses lead to the increase in [ROS]_i_ and [Ca^2+^]_i_. The increased [ROS]_i_ and [Ca^2+^]_i_ cause pulmonary vasoconstriction and vascular remodeling, leading to pulmonary hypertension. These signaling pathways are also nicely regulated by the nuclear hormone receptor REV-ERBα- or REV-ERBβ-regulated transcription factor BMAL1- and CLOCK-controlled circadian access.

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
