# Peer review of "Mitochondrial Dynamics in Pulmonary Hypertension"

_biomedicines, 2023, doi:10.3390/biomedicines12010053_

Round 1
Reviewer 1 Report
Comments and Suggestions for Authors
In this manuscript, the role of mitochondria in pulmonary vascular disease is reviewed. While this review highlights many interesting areas for future investigation in the lung, much of the described material linking mitochondria to tissue dysfunction focuses on responses in other tissues, i.e. skeletal muscle, neurons. The authors describe important potential relationships that have not been highlighted in previous reviews, however it seems that topics are also somewhat disorganized.
Major
1. It would be beneficial to include more specific examples of pulmonary arterial mitochondrial signaling as opposed to other tissues. Other vascular mitochondria signaling could also be considered in a greater capacity as opposed to the focus on other tissues such as skeletal muscle.
2. The authors provide a nice overview of how the signaling of the circadian clock can regulate mitochondrial function in a variety of tissues. However, the link between circadian clock and pulmonary vascular disease is somewhat tenuous in section 3. This is then discussed in section 5. The gap between these 2 sections makes this story very disjointed and it is suggested that these sections are either combined or reorganized to place section 5 immediately after section 3.
The above is the most obvious example, but throughout the manuscript, this reviewer found many topics to be disjointed and challenging to follow the progression of the story during the initial readthrough. I freely admit that this is a subjective opinion, but hope the authors will consider revisions which make their timely discussion of mitochondrial dysfunction in pulmonary vascular disease easier to follow. An additional example relates to discussion of NOX4 (page 7). The 1st paragraph discusses NOX 4 in pulmonary hypertension without linking to mitochondria, the next paragraph then moves to PCK epsilon, then the topic turns back to NOX 4 and mitochondria in the 3rd paragraph, and then returns to PCK epsilon in the 4th paragraph. Staying on topic and completing each sub-story individually will enhance the presentation of the authors discussion.
3. The authors highlight the role of PKC epsilon in mitochondrial function in pulmonary hypertension and hypoxia. PKC beta has also been found to contribute to mitochondrial ROS production (PMID: 32048876) and regulate pulmonary vascular reactivity in response to intermittent hypoxia and inclusion of such discussion may be considered for inclusion in this review.
Comments on the Quality of English LanguageA handful to typos readily cleaned up by editing. For example:
1. Page 5, line 16. Missing period and space between “BMAL” and “Several”.
2. Section 6 title, “leading to PH”?
Author Response
In this manuscript, the role of mitochondria in pulmonary vascular disease is reviewed. While this review highlights many interesting areas for future investigation in the lung, much of the described material linking mitochondria to tissue dysfunction focuses on responses in other tissues, i.e. skeletal muscle, neurons. The authors describe important potential relationships that have not been highlighted in previous reviews, however it seems that topics are also somewhat disorganized.
Thank you very much for your kindest overall appreciation of our current manuscript.
Major
- It would be beneficial to include more specific examples of pulmonary arterial mitochondrial signaling as opposed to other tissues. Other vascular mitochondria signaling could also be considered in a greater capacity as opposed to the focus on other tissues such as skeletal muscle.
We appreciate the suggestion. We made some changes to the text to strengthen the discussion on lung tissues instead of other tissues. We added some information about PASMCs and removed some information about cardiomyocytes, neurons, and skeletal muscle cells that were not so relevant to the focus of this review.
- The authors provide a nice overview of how the signaling of the circadian clock can regulate mitochondrial function in a variety of tissues. However, the link between the circadian clock and pulmonary vascular disease is somewhat tenuous in section 3. This is then discussed in section 5. The gap between these 2 sections makes this story very disjointed and it is suggested that these sections are either combined or reorganized to place section 5 immediately after section 3.
It makes sense. We organize the sessions to bring information about the circadian cycle closer together and have greater reading fluency. We placed section 5 right after section 3 because both talk about the circadian cycle and make the reading better. Section 4, which talks about hypoxia, is now section 5.
The above is the most obvious example, but throughout the manuscript, this reviewer found many topics to be disjointed and challenging to follow the progression of the story during the initial read-through. I freely admit that this is a subjective opinion, but hope the authors will consider revisions which make their timely discussion of mitochondrial dysfunction in pulmonary vascular disease easier to follow. An additional example relates to discussion of NOX4 (page 7). The 1st paragraph discusses NOX 4 in pulmonary hypertension without linking to mitochondria, the next paragraph then moves to PCK epsilon, then the topic turns back to NOX 4 and mitochondria in the 3rd paragraph, and then returns to PCK epsilon in the 4th paragraph. Staying on topic and completing each sub-story individually will enhance the presentation of the author’s discussion.
We appreciate the suggestion, the layout of the paragraphs was better, leaving NOX information together and then introducing the PKC.
- The authors highlight the role of PKC epsilon in mitochondrial function in pulmonary hypertension and hypoxia. PKC beta has also been found to contribute to mitochondrial ROS production (PMID: 32048876) and regulate pulmonary vascular reactivity in response to intermittent hypoxia and inclusion of such discussion may be considered for inclusion in this review.
The suggested article has been cited, indicating that PKCβ also contributed to mitochondrial ROS production and thus regulates pulmonary vascular reactivity in response to intermittent hypoxia. Thank you very much.
Comments on the Quality of English Language
A handful to typos readily cleaned up by editing. For example:
- Page 5, line 16. Missing period and space between “BMAL” and “Several”.
- Section 6 title, “leading to PH”?
We have changed all these issues, thanks.
Reviewer 2 Report
Comments and Suggestions for Authors
The authors reviewed the mitochondrial dynamics and viable circadian clock in hypoxic pulmonary hypertension.
It is well known that hypoxia/chronic stress induces mitochondrial dysfunction, which is significantly associated with pulmonary hypertension due to exposure to hypoxia.
The manuscript is well-written and summarizes relevant information with updated references. Nevertheless, the authors focused only on pulmonary hypertension... in this case, the title "Mitochondrial dynamics in pulmonary vascular pathologies" is unsuitable.
I suggest changing the "Vascular pathologies" for "Pulmonary hypertension"
Author Response
The authors reviewed the mitochondrial dynamics and viable circadian clock in hypoxic pulmonary hypertension.
It is well known that hypoxia/chronic stress induces mitochondrial dysfunction, which is significantly associated with pulmonary hypertension due to exposure to hypoxia.
The manuscript is well-written and summarizes relevant information with updated references. Nevertheless, the authors focused only on pulmonary hypertension... in this case, the title "Mitochondrial dynamics in pulmonary vascular pathologies" is unsuitable.
I suggest changing the "Vascular pathologies" for "Pulmonary hypertension"
We have taken your excellent suggestion; thus, the current title is "Mitochondrial Dynamics in pulmonary hypertension”.
Reviewer 3 Report
Comments and Suggestions for Authors
I commend the authors on their comprehensive review focusing on mitochondrial dynamics in pulmonary vascular pathologies, particularly in relation to pulmonary hypertension (PH). The manuscript effectively highlights the critical role of mitochondrial fusion and fission processes, the impact of circadian rhythms, and the influence of external factors such as hypoxia and nicotine. This work provides valuable insights into the complex interplay between these factors and their contribution to disease progression.
However, I recommend a more detailed exploration of the molecular mechanisms underlying mitochondrial dynamics and their dysregulation in disease states. Additionally, while the effects of nicotine on mitochondrial function are well-presented, expanding on potential therapeutic interventions for nicotine-induced mitochondrial dysfunction could enhance the manuscript's clinical relevance. The integration of these complex topics could be streamlined for clearer narrative cohesion.
Overall, the manuscript makes a significant contribution to our understanding of mitochondrial biology in pulmonary vascular diseases and opens avenues for future research and therapeutic development.
- Scope and Relevance: The manuscript provides an insightful overview of the role of mitochondrial dynamics in pulmonary vascular biology and disease processes, particularly in the context of pulmonary hypertension (PH). The focus on the interplay between mitochondrial fusion-fission dynamics, circadian rhythms, and the impacts of hypoxia and nicotine is particularly relevant and timely, given the increasing prevalence of respiratory diseases and lifestyle-related risk factors​​​​​​.
- Mitochondrial Dynamics and Disease Processes: The manuscript does an excellent job of detailing how alterations in mitochondrial dynamics (fusion and fission) contribute to the pathogenesis of pulmonary vascular diseases. However, the paper could benefit from more explicit discussion of the molecular mechanisms underlying these processes. While the roles of specific proteins like DRP1 and MFN1/2 are mentioned, a deeper dive into how these proteins interact and how their dysregulation contributes to disease could enhance the reader's understanding​​.
- Circadian Rhythms and Mitochondrial Function: The discussion on the influence of the circadian clock on mitochondrial function is intriguing. It's commendable how the manuscript links circadian disruptions with mitochondrial dysfunction and subsequent disease pathology. This aspect could be expanded to explore potential therapeutic avenues targeting circadian rhythm regulation as a means to mitigate mitochondrial dysfunctions and associated diseases​​.
- Nicotine’s Impact on Mitochondrial Function: The sections addressing nicotine's impact on mitochondrial dynamics and its contribution to diseases like COPD and PH are particularly insightful. The manuscript effectively ties together the impact of nicotine on mitochondrial fission, the associated increase in ROS production, and the downstream pathological consequences. However, there seems to be room for further discussion on the potential for therapeutic intervention in nicotine-induced mitochondrial dysfunction, which could be of significant clinical relevance​​​​.
- Clinical Implications and Therapeutic Potential: The manuscript provides a comprehensive overview of the pathophysiological implications of altered mitochondrial dynamics. However, it could benefit from a more detailed discussion on the translational and clinical implications of these findings. Specifically, exploring potential therapeutic strategies that target mitochondrial dynamics, circadian rhythm regulation, or specific pathways affected by hypoxia and nicotine could provide a more rounded perspective and offer insights into future research directions​​.
- Integration of Concepts: The manuscript does well in integrating various complex concepts, but it might benefit from a clearer, more streamlined narrative that connects these concepts more cohesively. This would help in better conveying the complexity of the interplay between mitochondrial dynamics, circadian rhythms, and external factors like hypoxia and nicotine.
Overall, the manuscript provides a comprehensive and detailed exploration of the role of mitochondrial dynamics in pulmonary vascular pathologies. It highlights critical areas for future research and potential therapeutic strategies, although there is scope for further exploration of molecular mechanisms and therapeutic implications.
Author Response
I commend the authors on their comprehensive review focusing on mitochondrial dynamics in pulmonary vascular pathologies, particularly in relation to pulmonary hypertension (PH). The manuscript effectively highlights the critical role of mitochondrial fusion and fission processes, the impact of circadian rhythms, and the influence of external factors such as hypoxia and nicotine. This work provides valuable insights into the complex interplay between these factors and their contribution to disease progression. However, I recommend a more detailed exploration of the molecular mechanisms underlying mitochondrial dynamics and their dysregulation in disease states. Additionally, while the effects of nicotine on mitochondrial function are well-presented, expanding on potential therapeutic interventions for nicotine-induced mitochondrial dysfunction could enhance the manuscript's clinical relevance. The integration of these complex topics could be streamlined for clearer narrative cohesion. Overall, the manuscript makes a significant contribution to our understanding of mitochondrial biology in pulmonary vascular diseases and opens avenues for future research and therapeutic development.
We appreciate your valuable suggestions. As such, we have included the more detailed molecular mechanisms underlying mitochondrial dynamics and their dysregulation in disease states. Also, we have expanded the descriptions of potential therapeutic interventions for nicotine-induced mitochondrial dysfunction and its clinical relevance.
- Scope and Relevance: The manuscript provides an insightful overview of the role of mitochondrial dynamics in pulmonary vascular biology and disease processes, particularly in the context of pulmonary hypertension (PH). The focus on the interplay between mitochondrial fusion-fission dynamics, circadian rhythms, and the impacts of hypoxia and nicotine is particularly relevant and timely, given the increasing prevalence of respiratory diseases and lifestyle-related risk factors​​​​​​.
Thank you for your appreciated compliments.
- Mitochondrial Dynamics and Disease Processes: The manuscript does an excellent job of detailing how alterations in mitochondrial dynamics (fusion and fission) contribute to the pathogenesis of pulmonary vascular diseases. However, the paper could benefit from more explicit discussion of the molecular mechanisms underlying these processes. While the roles of specific proteins like DRP1 and MFN1/2 are mentioned, a deeper dive into how these proteins interact and how their dysregulation contributes to disease could enhance the reader's understanding.
We have included a more explicit discussion of the molecular mechanisms underlying the role of mitochondrial dynamics in the pathogenesis of pulmonary vascular diseases.
- Circadian Rhythms and Mitochondrial Function: The discussion on the influence of the circadian clock on mitochondrial function is intriguing. It's commendable how the manuscript links circadian disruptions with mitochondrial dysfunction and subsequent disease pathology. This aspect could be expanded to explore potential therapeutic avenues targeting circadian rhythm regulation as a means to mitigate mitochondrial dysfunctions and associated diseases.
More discussion on the links of circadian disruptions with mitochondrial dysfunction and subsequent disease pathology has been added.
- Nicotine’s Impact on Mitochondrial Function: The sections addressing nicotine's impact on mitochondrial dynamics and its contribution to diseases like COPD and PH are particularly insightful. The manuscript effectively ties together the impact of nicotine on mitochondrial fission, the associated increase in ROS production, and the downstream pathological consequences. However, there seems to be room for further discussion on the potential for therapeutic intervention in nicotine-induced mitochondrial dysfunction, which could be of significant clinical relevance​​​​.
Further discussion on the potential for therapeutic intervention in nicotine-induced mitochondrial dysfunction has been included.
- Clinical Implications and Therapeutic Potential: The manuscript provides a comprehensive overview of the pathophysiological implications of altered mitochondrial dynamics. However, it could benefit from a more detailed discussion on the translational and clinical implications of these findings. Specifically, exploring potential therapeutic strategies that target mitochondrial dynamics, circadian rhythm regulation, or specific pathways affected by hypoxia and nicotine could provide a more rounded perspective and offer insights into future research directions.
Exploring potential therapeutic strategies that target mitochondrial dynamics, circadian rhythm regulation, or specific pathways affected by hypoxia and nicotine has been provided to further clarify a more rounded perspective and offer insights into future research directions. As already mentioned, including new information regarding therapeutic alternatives
- Integration of Concepts: The manuscript does well in integrating various complex concepts, but it might benefit from a clearer, more streamlined narrative that connects these concepts more cohesively. This would help in better conveying the complexity of the interplay between mitochondrial dynamics, circadian rhythms, and external factors like hypoxia and nicotine. Overall, the manuscript provides a comprehensive and detailed exploration of the role of mitochondrial dynamics in pulmonary vascular pathologies. It highlights critical areas for future research and potential therapeutic strategies, although there is scope for further exploration of molecular mechanisms and therapeutic implications.
A clearer, more streamlined narrative that connects mitochondrial dynamics to circadian rhythms to external factors like hypoxia and nicotine has been included.
Round 2
Reviewer 1 Report
Comments and Suggestions for Authors
Comments have been adequately addressed.
Comments on the Quality of English LanguageMinor phrasing issues.
Reviewer 3 Report
Comments and Suggestions for Authors
The authors accommodated my queries. I am satisfied with the revision.